# A Liquid Chromatography Tandem Mass Spectrometry Method for the Simultaneous Estimation of the Dopamine Receptor Antagonist LE300 and Its N-methyl Metabolite in Plasma: Application to a Pharmacokinetic Study

**DOI:** 10.3390/molecules28041553

**Published:** 2023-02-06

**Authors:** Mohamed M. Hefnawy, Mohamed W. Attwa, Adeeba A. Alzamil, Manal A. El-Gendy, Adel S. El-Azab, Yousef A. Bin Jardan, Ali A. El-Gamal

**Affiliations:** 1Department of Pharmaceutical Chemistry, College of Pharmacy, King Saud University, Riyadh 11451, Saudi Arabia; 2Department of Analytical Chemistry, Faculty of Pharmacy, Mansoura University, Mansoura 35516, Egypt; 3Department of Pharmaceutics, College of Pharmacy, King Saud University, Riyadh 11451, Saudi Arabia; 4Department of Pharmacognosy, College of Pharmacy, King Saud University, Riyadh 11451, Saudi Arabia

**Keywords:** LC-MS/MS, LE300, N-methyl metabolite, pharmacokinetic study

## Abstract

LE300 is a novel dopamine receptor antagonist used to treat cocaine addiction. In the current study, a sensitive and fast liquid chromatography–tandem mass spectrometry (LC-MS/MS) has been established and validated for the simultaneous analysis of LE300 and its *N*-methyl metabolite, MLE300, in rat plasma with an application in a pharmacokinetic study. The chromatographic elution of LE300, MLE300, and Ponatinib (IS, internal standard), was carried out on a 50 mm C_18_ analytical column (ID: 2.1 mm and particle size: 1.8 μm) maintained at 22 ± 2 °C. The run time was 5 min at a flow rate of 0.3 mL/min. The mobile phase consisted of 42% aqueous solvent (10 mM ammonium formate, pH: 4.2 with formic acid) and 58% organic solvent (acetonitrile). Plasma samples were pretreated using protein precipitation with acetonitrile. The electrospray ionization (ESI) source was used to generate an ion-utilizing positive mode. A multiple reaction monitoring mass analyzer mode was utilized for the quantification of analytes. The linearity of the calibration curves in rat plasma ranged from 1 to 200 ng/mL (r^2^ = 0.9997) and from 2 to 200 ng/mL (r^2^ = 0.9984) for LE300 and MLE300, respectively. The lower limits of detection (LLOD) were 0.3 ng/mL and 0.7 ng/mL in rat plasma for LE300 and MLE300, respectively. Accuracy (RE%) ranged from −1.71% to −0.07% and −4.18% to −1.48% (inter-day), and from −3.3% to −1.47% and −4.89% to −2.15% (intra-day) for LE300 and MLE300, respectively. The precision (RSD%) was less than 2.43% and 1.77% for the inter-day, and 2.77% and 1.73% for intra-day of LE300 and MLE300, respectively. These results are in agreement with FDA guidelines. The developed LC-MS/MS method was applied in a pharmacokinetic study in Wistar rats. T_max_ and C_max_ were 2 h and 151.12 ± 12.5 ng/mL for LE300, and 3 h and 170.4 ± 23.3 ng/mL for MLE300.

## 1. Introduction

LE300 (7-methyl-6,7,8,9,14,15-hexahydro-5Hbenz[d]indolo[2,3-g]azecine) (Figure 1A) is a member of a dopamine receptor ligand class combining structural core elements of dopamine and serotonin. It is an antagonist for the azecine-type dopamine (D) receptor and is characterized by higher affinities for dopamine receptors and exclusive selectivity profiles against D_1_-like receptors [1,2,3]. D_1_ is a brain neurotransmitter that is thought to be involved in several physiological functions, including cognition, behavior, locomotion, motivation, and learning. Many neuropsychiatric disorders have been linked to dopaminergic system dysfunction [4].

Methylation is a relatively minor conjugation pathway in drug metabolism, but it is very important in the biosynthesis of endogenous compounds, such as epinephrine and melatonin; and in the catabolism of biogenic amines, such as dopamine, serotonin, and histamine. In many cases, this conjugation results in compounds with decreased biological activity [5]. Methylation is a two-step process. First, the methyl-transferring coenzyme, S-adenosylmethionine (SAM), is biosynthesized mostly from methionine in a reaction catalyzed by methionine adenosyltransferase. Then, the SAM is utilized in the transfer of the activated methyl group to the acceptor molecule (LE300), as shown in Figure 2. Heterocyclic nitrogen atoms molecule such as LE300 are susceptible to N-methylation in human results in compounds with decreased biological activity.

Upon a literature review, we found three analytical methods to quantify LE300 and MLE300 [6,7,8]. When using the spectrofluorimetric method [6], the linearity ranges from 5 to 100 ng/mL, which is less sensitive than our method and involves a longer sample preparation time. In the HPLC–Fluorescence method [7], the linearity ranges from 4 to 500 ng/mL, which is less sensitive than our method and consumes a lot of solvents, as the flow rate is 1 mL/min. In the microemulsion liquid chromatography method [8], the linearity ranges from 10 to 400 ng/mL for LE-300, which is much lower in comparison to our developed methodology. Hence, our established method is more sensitive, faster, and consumes less solvent than previously reported methods. No LC-MS/MS method has been reported for the analysis of LE300 and/or its N-methyl metabolite. This work describes, for the first time, the establishment and validation of an LC-MS/MS method for the simultaneous quantification of LE300 and its metabolite, MLE300 (Figure 1B), in rat plasma using a 50 mm C_18_ column using Ponatinib (Figure 1C) as the internal standard.

The established bio-analytical method was validated following FDA guidelines [9]. Analytical method validation is necessary to demonstrate the suitability for any desired pharmacokinetic application. The method has to produce precise, reproducible, and accurate results; this is necessary for bioavailability, pharmacodynamics, pharmacokinetics, bioequivalence, or toxicological studies, since such methods are used in analyte quantification of various biological matrices, such as plasma or urine [10,11,12,13].

## 2. Results and Discussion

### 2.1. Chromatography and Mass Spectrometry Parameters

Numerous trials were performed to obtain the maximum mass response by adjusting all chromatographic parameters so as to enhance resolution and sensitivity. The pH of the aqueous portion of the elution system can improve the analytes’ ionization and adjust peak shape. Various mobile phase compositions were tested. The adjusted condition was as follows: 42% ACN, 58% 10 mM ammonium formate in water (pH~4.2). LE300, MLE300, and Ponatinib (IS) elution times were 2.0, 3.0, and 3.8 min, respectively, using the optimized chromatographic conditions. We investigated the use of different internal standards such as pemigatinib, nateglinide, repaglinide, chloroquine, and hydroxychloroquine, but such internal standards either showed poor peaks or led to overlapping with LE300 or MLE300. Ponatinib was selected as the method’s IS, whereas it has a higher extraction recovery (≥99%) and performance characteristics for LE300 and MLE300 [9]. Sample processing by liquid–liquid extraction and protein precipitation using different solvents was tried. It was found that protein precipitation utilizing acetonitrile is the optimum method with regard to simplicity, affordability and easier sample processing. A 5 min run time was enough to achieve complete elution of the three analytes (LE300, MLE300, and IS) without carryover being noticed in a blank rat plasma sample. Figure 3 shows overlayed chromatograms with good resolution for the calibration levels of LE300, MLE300, and IS in rat plasma.

Mass spectrometry chromatographic parameters were optimized to increase the ionization of the molecular ions of the parent and its PIs for LE300, MLE300, and IS. MRM mode was utilized in this work to erase any expected interferences and increase the method’s sensitivity. The full mass scans spectra of analytes composed of one molecular ion were detected at *m*/*z* 291, *m*/*z* 305, and *m*/*z* 533 for LE300, MLE300, and IS, respectively. A positive PI scan for LE300 (*m*/*z* 292) generated major product ions at *m*/*z* 160 and *m*/*z* 246. A positive PI scan for MLE300 (*m*/*z* 305) generated major product ions of [M + H]^+^ at *m*/*z* 158 and *m*/*z* 248. A positive PI scan for Ponatinib (*m*/*z* 533) generated a two-fragment ion of [M + H]^+^ at *m*/*z* 260 and m/z 433. These ions were chosen for quantification of LE300, MLE300, and IS using MRM mode (Figure 4, Table 1).

### 2.2. Method Validation

#### 2.2.1. Selectivity

The analytical method selectivity was validated by a comparison between the MRM chromatograms of drug-free rat plasma and the equivalent spiked samples at LLOQ levels after injection of MLE300 and LE300. The retention times of LE300, MLE300, and IS were 2.0, 3.0, and 3.8 min, respectively. No noticeable endogenous material interference was seen in the MRM chromatograms of the drug-free plasma at the elution times of the two analytes and IS. The carryover in the blank sample was less than 20% of LLOQ for LE300 and MLE300, and less than 5% of the response for IS after injection of the upper limit of quantification (ULOQ) of the calibration curve [9].

#### 2.2.2. Calibration Curve

The proposed method was reliable and sensitive for LE300 and MLE300 determination in rat plasma. The least-square statistical method was used for the linear regression analysis of the outcomes. The linearity of the method ranged from 1 to 200, and from 2 to 200 ng/mL, with a correlation coefficient (r^2^) >0.9997 and 0.9984 in LE300 and MLE300, respectively, as seen in the calibration curve. The regression equations of LE300 and MLE300 calibration curves were y = 0.0034x + 0.0179 and y = 0.0129x + 0.0238, respectively. LLODs were found to be 0.3 and 0.7 ng/mL in rat plasma for LE300 and MLE300, respectively, confirming the applicability of the developed assay for the quantification of trace concentrations LE300 and MLE300 in plasma. The high r^2^ value showed good linearity. The low values of standard deviations of the slope and the intercept revealed the calibration standard points’ validity, and those points were used to establish the calibration curve (Table 2).

The relative SD values of each calibration standard level (six repeats) did not exceed 2.39% and 2.20% for LE300 and MLE300, respectively. Calibration standards and QC samples of LE300 and MLE300 in rat plasma samples (eleven levels) were back-calculated in order to find the best methodology performance. The accuracy and precision for LE300 and MLE300 in rat plasma samples ranged from 0.056 to 2.199%, from 0.865% to 2.39%, from −1.64% to −0.87%, and from −3.819% to −1.1%, respectively. The average LE300, MLE300, and Ponatinib (IS) recoveries were 99.49 ± 1.53%, 99.03 ± 1.52% and 99.49 ± 1.53% in rat plasma samples, respectively (Table 3).

#### 2.2.3. Accuracy and Precision

The analytical method reproducibility was confirmed using intra- and inter-day accuracy and precision measurements at four concentrations of QC samples (LLOQ, LQC, MQC, and HQC) in six replicates for LE300 and MLE300, respectively. Precision and accuracy were expressed by percentage relative SD (% RSD) and percentage error (% error) values, respectively. As mentioned in Table 4, the accuracies (RE%) ranged from −7.00% to −1.94%, and −6.50% to −2.62% for the inter-day, and from −6.00% to −1.70% and −5.50% to −2.39% for the intra-day of LE300 and MLE300, respectively. The corresponding precisions (RSD %) were less than 2.86% and 3.17%, and less than 2.62% and 2.18% for LE300 and MLE300, respectively. These values met the acceptance criteria of the guidelines; LLOQ within 20% and the other QCs within 15% [9].

#### 2.2.4. Matrix Effects and Extraction Recovery

The extraction recoveries of LE300 and MLE300 QC samples from rat plasma are summarized in Table 5. The outcomes were reproducible, reliable, and accurate. To confirm the lack of matrix effect on the LE300 and MLE300 analysis, six different batches of rat plasma samples were extracted and spiked with 3 ng/mL LE300 (LQC), 75 ng/mL LE300 (MQC), and 175 ng/mL LE300 (HQC) for LE300, 6 ng/mL MLE300 (LQC), 90 ng/mL MLE300 (MQC), and 180 ng/mL MLE300 (HQC) for MLE300, and IS (20 ng/mL) to form set 1. Set 2 was prepared in a similar way, with similar concentrations of LE300 and MLE300 and IS solubilized in the mobile phase. For the estimation of the matrix effect, the average peak area ratio of set 1/set 2 × 100 was calculated. The rat plasma samples containing LE300 and MLE300 had values of 98.67 ± 2.5% and 98.29 ± 1.62%, respectively. The mean RSD was 1.3–1.9% and 0.94–1.42% for LE300 and MLE300 in rat plasma samples, respectively. From the previous results, we conclude that rat plasma has no noticeable effect on the ionization of LE300, MLE300, and IS.

#### 2.2.5. Stability

LE300 and MLE300 stabilities in stock preparations and in rat plasma were measured using laboratory conditions that samples might be subjected to before analysis. LE300 and MLE300 exhibited perfect stability in stock preparations after being kept at −80 °C for 28 days. The stability values ranged from 98.2 to 99.1% and from 98.4 to 99.5% for LE300 and MLE300, respectively, in rat plasma samples (as shown in Table 6). There was no observed loss of the analytes after short-term storage, autosampler storage, three freeze–thaw cycles, and long-term storage. The results indicate that perfect stability of LE300 and MLE300 had been achieved.

### 2.3. Pharmacokinetic Study

The established LC-MS/MS methodology was used in a pharmacokinetic study of LE300 in rats. The concentrations of LE300 and MLE300 in rat plasma were determined individually at different time intervals after oral administration of 10 mg/kg of LE300 [14]. The typical MRM chromatograms of rat plasma 2.0 h after oral administration are presented in Figure 5. A plasma concentration–time curve is shown in Figure 6, and the PK parameters are summarized in Table 7. The mean values of T_max_ and C _max_ were 2 h and 150.15 ± 17.68 ng/mL for LE300, and 3 h and 170.4 ± 23.3 ng/mL for MLE300, respectively. The AUC_0–24_ for LE300 was found to be 523.61 ± 24.26 ng·h/mL. The AUC_0–∞_ for LE300 was found to be 576.82 ± 22.36 ng·h/mL. The elimination half-life (t_1/2kel_) for LE300 was found to be 7.38 ± 0.17. Samples obtained from the control animals that were treated with a vehicle-free drug showed no peaks at LE300 and MLE300 elution times. This indicates that N-methylation may be the metabolic reaction rather than any other pathway.

## 3. Material and Methods

### 3.1. Materials

All chemicals used in the current study are mentioned in detail in Table 8. Maintenance of rats was performed following the Animal Care Center guidelines of the College of Pharmacy (King Saud University, Saudi Arabia). The protocol that was used for animal experiments was approved by the Institutional Review Board. The animal research ethics committee approved this research (approval number KSU-SE-18-19). After the experiment was complete, the animals were left for a wash-out period (i.e., two weeks), then they were either used in other animal studies in our laboratory or euthanized.

### 3.2. LC–MS/MS Analysis

All chromatographic LC-MS/MS parameters were adjusted to achieve fast elution with high resolution (Table 9). Quantification of analytes was conducted using MRM mode (Figure 4).

### 3.3. Calibration Standard Solutions and QC Sample Preparation

LE300 and MLE300 stock solutions were prepared separately in a mixture of methanol and water (1:1) at a concentration of 1 mg/mL. Ponatinib (IS) stock solution was prepared in DMSO at 1 mg/mL. All solutions were kept at −20 °C for stabilization. Successive working solutions of LE300 and MLE300 were additionally obtained through dilution using ultrapure water at concentrations of 0.05, 0.5, 5, and 20 µg/mL. A working solution of IS was prepared in ultrapure water at a concentration of 20 ng/mL. Two calibrators at concentrations ranged from 1 to 200 ng/mL and 2 to 200 ng/mL for LE300 and MLE300, respectively, were prepared in blank rat plasma from the intermediate solutions. Different quality control samples for LE300 at 1.0 ng/mL for the LLOQ, 3.0 ng/mL for the QC sample at low concentration, (LOQ), 75 ng/mL for the QC sample at mid concentration (MQC), and 175 ng/mL for the QC sample at high concentration (HQC) were prepared by spiking an appropriate volume of the intermediate solutions with blank rat plasma. Similarly, different quality control samples for MLE300 at 2.0, 6.0, 90, and 180 ng/mL for LLOQ, LOQ, MQC, and HQC were prepared. The peak area ratios of LE300 and MLE300 to IS were treated to obtain the calibration curve of each drug. Alternatively, the corresponding regression equation was derived.

### 3.4. Sample Preparation

Fifty µL of each calibrator and quality controls plasma samples was transferred to 2.0 mL disposable polypropylene centrifuge tubes. Fifty µL of IS (20 ng/mL) was added equally to each tube, then diluted to 750 µL with ultrapure water and vortexed for 0.5 min. An amount of 500 µL of acetonitrile was added to the spiked plasma samples to precipitate the plasma proteins and mixed for 1 min. The tubes were subsequently vortexed for 1 min and centrifuged at 10,000 rpm at 5 °C for 12 min. Next, 500 µL of supernatants were collected to clean tubes, then evaporated to dryness under nitrogen. Residues were reconstituted using 500 µL of mobile phase, 10 mM ammonium formate, acetonitrile (42:58 *v/v*), and then filtered through a 0.22-μm ChromTech Nylon Membrane Filter (ChromTech, Kent, UK). Filtrates were loaded in HPLC vials in the well plate sampler tray, and 5 μL of the eluent was injected into the chromatographic LC-MS/MS system. Rat plasma samples from treated rats were prepared in a similar way as previously mentioned.

### 3.5. Method Validation

The guidelines for bio-analytical method validation of the Food and Drug Administration (US-FDA) were followed for validation purposes [9]. Validation of the developed method in the rat plasma was carried out using specificity, sensitivity, linearity, precision, accuracy, extraction recovery, stability, dilution integrity, and matrix effect [14,15,16,17].

### 3.6. Application to Pharmacokinetic Studies

All animal experiments were conducted according to the standards set forth in the experimental animals use and care guidelines by the National Institute of Health (NIH) [18]. Approval for the study was granted by the Animal Ethics Committee of the Pharmacology Department at the College of Pharmacy in King Saud University (Saudi Arabia; number KSU-SE-18-19). Six healthy Wistar male rats (250 ± 30 g) were kindly gifted from the Experimental Animal Care Center. The animals were maintained in a well-ventilated room inside cages under a 12 h day/night cycle at specific conditions (40–60% relative humidity and 24–27 °C temperature). All the rats had free access to water, while the diet was stopped for 12 h prior to drug loading. The rats were kept for one week in a laboratory before the experiments were performed. On the day of experiments, rats were treated by gavage administration with a single oral dose of 10 mg/kg LE300 dissolved in 1% DMSO/saline [19]. Blood samples (300 μL) were collected into heparinized 1.5 mL polythene tubes containing ethylenediamine tetraacetic acid dipotassium (EDTA K_2_) (anticoagulant) before drug administration, and at 0.5, 1, 2, 4, 6, 8, 12, 18, and 24 h after oral administration of a single oral dose of 10 mg/kg LE300 [14]. The samples were directly centrifuged at 3000 rpm for 10 min at 4 °C. The plasma obtained was stored at −80 °C until analysis. The same method of extraction described under calibration standards preparation (2.4.) was used for sample preparation. The PK parameters of LE300 and MLE300, such as C_max_, T_max_, t1/_2kel_, AUC_0–24_, and AUC_0–∞_, were calculated by fitting the data to a non-compartmental analysis (NCA) model with PK Solver Add-In software [20].

## 4. Conclusions

A sensitive, rapid, and simple LC-MS/MS methodology was established and validated to estimate LE300 and MLE300 in rat plasma. The developed method exhibited a linear range from 1 to 200 and 2 to 200 ng/mL with LLOQ values less than 1 and 2 ng/mL for LE300 and MLE300, respectively. The elution time was fast (5 min) with lower solvent consumption. Estimating LE300 and MLE300 samples on the same day and/or on consecutive days demonstrated acceptable levels of precision and accuracy of the established protocol. The developed LC-MS/MS methodology was also characterized by minimal sample preparation, and good accuracy and precision. Therefore, this method is useful for the toxicological and therapeutic monitoring of LE300 in clinical practice. The developed method was used successfully for the pharmacokinetic study of LE300 in rats.

## Figures and Tables

**Figure 1 molecules-28-01553-f001:**
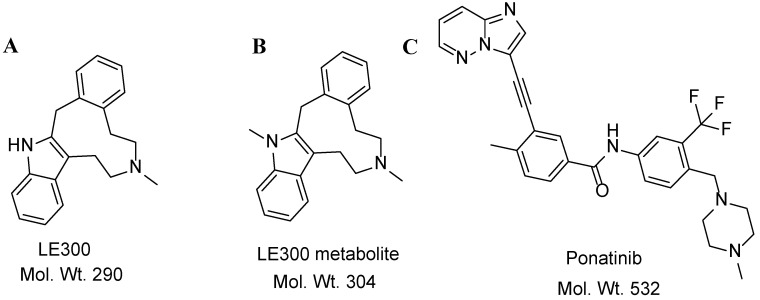
Chemical structures of LE300 (**A**), MLE300 (**B**), and Ponatinib (**C**).

**Figure 2 molecules-28-01553-f002:**
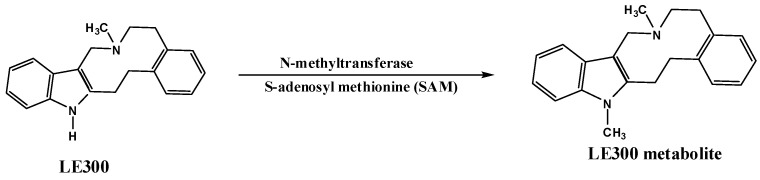
Metabolism pathway of LE300.

**Figure 3 molecules-28-01553-f003:**
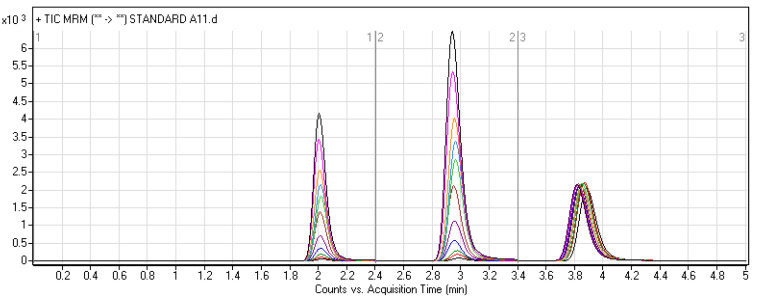
Overlayed TIC chromatograms of MRM of LE300 (1–200 ng/mL), MLE300 (2–200 ng/mL), and IS (20 ng/mL).

**Figure 4 molecules-28-01553-f004:**
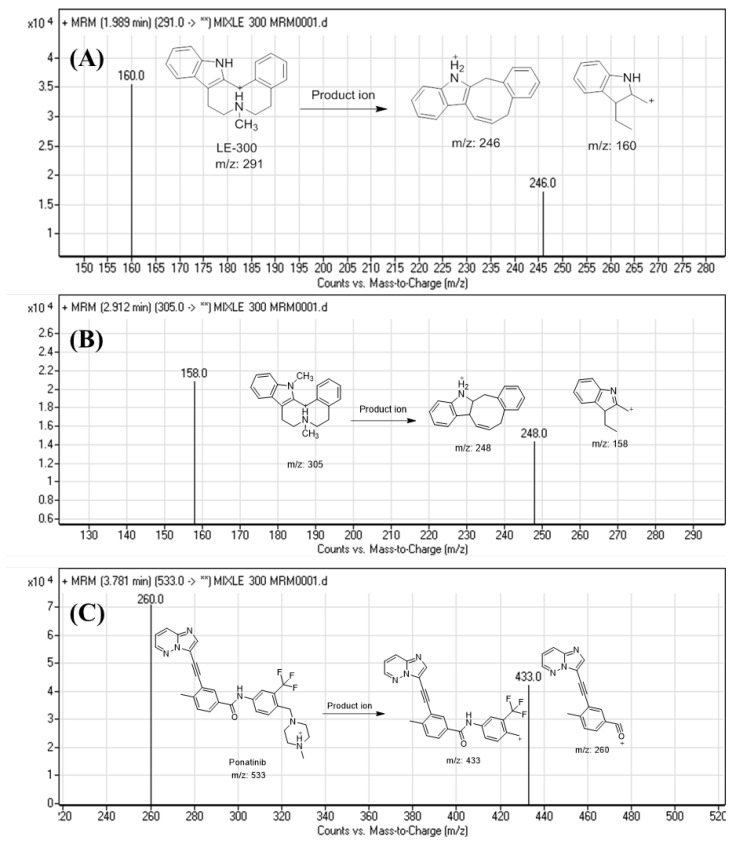
MRM spectra and the proposed PIs of (**A**) LE300, (**B**) MLE300, and (**C**) IS.

**Figure 5 molecules-28-01553-f005:**
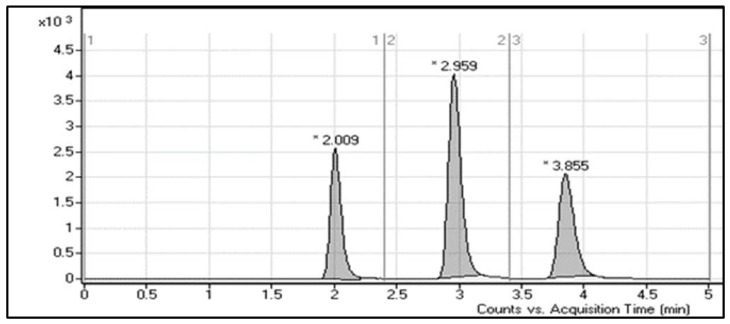
Typical multiple reaction monitoring (MRM) chromatograms for in vivo rat plasma sample 2.0 h after oral (gavage) administration of 10 mg/kg LE300; LE300 (2.01 min), KLE300 (2.96 min) and IS (3.86 min).

**Figure 6 molecules-28-01553-f006:**
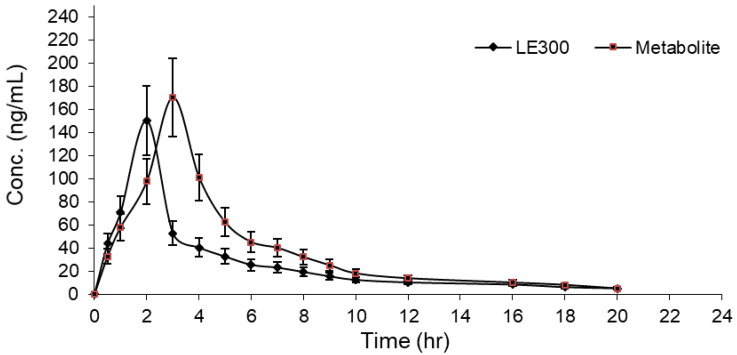
Concentration vs. time profile of LE300 and MLE300 in rat plasma after LE300 (10 mg/kg) oral administration. Each data represents the mean ± SD of six rats.

**Table 1 molecules-28-01553-t001:** MRM mode parameters.

Molecules	Elution Time (min)	Ion Mode	MRM Transitions (*m*/*z*)	Collision Energy (eV)
LE300	2.0 ± 0.1	+	291.0 > 160,246	20, 22
MLE300	3.0 ± 0.2	+	305 > 158,248	20, 18
Ponatinib (IS)	3.8 ± 0.1	+	533 > 260,433	18

**Table 2 molecules-28-01553-t002:** Validation parameters for the established LC-MS/MS method of LE300 and MLE300.

Parameters	LE300	MLE300
Linearity range (ng/mL)	1–200	2–200
Intercept (a)	1.79 × 10^−2^	2.38 × 10^−2^
Slope (b)	3.4 × 10^−3^	1.29 × 10^−3^
Correlation coefficient (r)	0.9997	0.9984
S_Y/N_ ^a^	1.01 × 10^−2^	2.38 × 10^−2^
S_a_ ^b^	5.63 × 10^−3^	3.43 × 10^−3^
S_b_ ^c^	3.12 × 10^−3^	1.43 × 10^−3^
LLOD (ng/mL)	0.3	0.7

^a^ SD of the residual, ^b^ SD of the intercept, ^c^ SD of the slope.

**Table 3 molecules-28-01553-t003:** Back-calculated data of LE300 and MLE300 of the calibration levels from rat plasma.

Nominal Conc. (ng/mL)		Mean ^a^		Precision (RSD%)		Accuracy (RE%)	
LE300	MLE300	LE300	MLE300	LE300	MLE300	LE300	MLE300
1	1	0.98 ± 0.12	0.98 ± 1.01	1.59	1.21	−2.33	−1.49
2	2	1.98 ± 0.07	2.01 ± 2.91	1.07	2.19	−1.10	−1.65
4	4	3.94 ± 0. 80	3.99 ± 1.23	1.98	0.69	−1.63	−1.04
8	8	7.89 ± 2.10	7.99 ± 0.19	0.87	0.06	−1.41	−1.08
16	16	15.64 ± 1.60	15.96 ± 3.20	1.59	0.99	−2.27	−1.38
32	32	30.78 ± 0.25	31.96 ± 1.92	1.58	0.61	−3.82	−1.57
60	60	59.17 ± 0.49	60.07 ± 1.63	2.39	0.53	−1.39	−1.48
80	80	78.97 ± 1.42	80.06 ± 1.65	1.42	0.51	−1.29	−1.49
100	100	98.48 ± 1.12	99.73 ± 1.79	1.42	0.80	−1.52	−1.54
120	120	117.94 ± 1.39	119.47 ± 1.80	1.62	0.73	−1.72	−1.01
160	160	156.45 ± 254	159.28 ± 2.37	1.62	0.45	−2.22	−1.49
200	200	195.62 ± 2.76	198.83 ± 1.26	1.41	0.45	−2.19	−0.87

^a^ Average of six replicates.

**Table 4 molecules-28-01553-t004:** Intra-day and inter-day precision and accuracy results of LE300 and MLE300 in rat plasma (Mean, *n* = 6).

	Actual Conc. (ng/mL)		Experimental (ng/mL)		RSD (%) ^a^		Accuracy (%) ^b^	
Analyte	LE300	MLE300	LE300	MLE300	LE300	MLE300	LE300	MLE300
Intra-day ^c^	1	2	0.94	1.89	2.12	2.51	−6.00	−5.50
	3	6	2.91	5.83	3.17	2.87	−3.00	−2.83
	75	90	73.72	86.98	1.37	1.86	−1.70	−3.35
	175	180	174.71	175.69	2.45	2.91	−1.70	−2.39
Inter-day ^c^	1	2	0.93	1.87	2.85	1.15	−7.00	−6.50
	3	6	2.89	5.83	2.56	1.97	−3.66	−2.83
	75	90	72.93	87.64	1.49	1.73	−2.76	−2.62
	175	180	171.60	174.25	2.86	2.18	−1.94	−3.19

^a^ % RSD: (SD/mean) × 100. ^b^ (Mean determined concentration/nominal concentration) × 100. ^c^ Mean based on *n* = 6.

**Table 5 molecules-28-01553-t005:** Recovery of LE300 and MLE300 QC samples in rat plasma.

Nominal Concentration (ng/mL)	LE300			MLE300			IS
	3 ng/mL	75 ng/mL	175 ng/mL	6 ng/mL	90 ng/mL	180 ng/mL	20 ng/mL
Mean ^a^	2.83	71.14	172.68	5.61	86.85	176.92	19.17
Recovery (%)	94.33	94.85	98.67	93.50	96.51	98.29	95.85
RSD%	1.76	1.37	1.48	2.13	1.85	1.94	1.96

^a^ Mean based on *n* = 6.

**Table 6 molecules-28-01553-t006:** Stability of LE300 and MLE300.

Analyte	Nominal Con. (ng/mL)	Freeze-Thaw Stability (3 Cycilic−80 °C)	Short-Team Stability (4 h at Room T)	Long-Team Stability (−80 °C for 28 d)	Autosampler Stability (24 h at 15 °C)
LE300	3	97.9 ± 1.5	98.5 ± 1.6	97.7 ± 1.4	98.5 ± 1.8
	175	99.2 ± 1.3	98.3 ± 1.8	99.2 ± 1.7	98.9 ± 1.3
MLE300	6	98.2 ± 1.4	97.9 ± 1.6	98.9 ± 1.4	99.3 ± 1.7
	180	99.5 ± 1.5	98.8 ± 1.8	99.2 ± 2.3	98.5 ± 1.7

**Table 7 molecules-28-01553-t007:** Pharmacokinetic parameters of LE300 and MLE300 after a single oral dose (10 mg/kg) in rat.

Parameters *	Unit	LE300	MLE300
AUC_0–24_ ^a^	ng·h/mL	523.61 ± 24.26	761.86 ± 34.29
AUC_0–∞_ ^b^	ng·h/mL	576.82 ± 22.36	789.54 ± 37.23
C_max_ ^c^	ng/mL	150.15 ± 17.68	170.40 ± 26.33
T_max_ ^d^	h	2.00 ± 0.18	3.00 ± 0.18
t_1/2kel_ ^e^	h	7.38 ± 0.17	4.56 ± 0.29
CL/F ^f^	L/h	0.017 ± 0.01	0.012 ± 0.01

* Data are described as the mean ± SD. ^a^ Area under the curve up to the last sampling time. ^b^ AUC extrapolated to infinity. ^c^ The maximum plasma concentration. ^d^ The time taken to reach the maximum plasma concentration. ^e^ Half-life in elimination phase. ^f^ Total clearance of the drug from plasma after oral administration.

**Table 8 molecules-28-01553-t008:** List of materials and chemicals.

Name	Source
N-methyl metabolite of LE300	Kindly gifted from Dr. J. Lehmann at Institut fur Pharmazie, Universitat Jena, Germany
Ponatinib	LC Laboratories (USA)
LE300, formic acid (HCOOH), acetonitrile (ACN) and ammonium formate	Sigma-Aldrich (USA)
HPLC grade water	Milli-Q plus purification instrument (Millipore, USA)
Wistar healthy male rats	The center for Experimental Animals at College of Pharmacy (KSU, Saudi Arabia)

**Table 9 molecules-28-01553-t009:** LC-MS/MS methodology.

Liquid Chromatography		Mass Spectrometer	
RRLC	Agilent 1200	Mass spectrometer	6410 Triple Quad of Agilent
Isocratic mobile phase	58% ACN	Ionization source	Positive ESI
	42% 10 mM ammonium formate (pH: 4.2 by addition of HCOOH)		Low-purity N_2_ gas as drying gas: Flow rate at 12 L/min with 60 psi pressure
	0.3 mL/min flow rate		
	5 μL injection volume		
Agilent eclipse plus C_18_ Column	50 mm L		Source T at 350 °C
	2.1 mm ID		Capillary voltage at 4000 V
	1.8 μm PS	Collision gas	High-purity N_2_ gas
	T: 22 ± 1 °C	Scan mode	MRM
Analyte	LE300	LE300 mass transitions FV: Fragmentor voltage CE: Collison energy	*m*/*z* 291→*m*/*z* 160 (FV is 145 V and CE: 20 eV)
			*m*/*z* 291→*m*/*z* 246 (FV is 140 V and CE: 22 eV)
Metabolite	N-methyl LE300 (MLE300)	MLE300 mass transitions FV: Fragmentor voltage CE: Collison energy	*m*/*z* 305→*m*/*z* 158 (FV is 140 V and CE: 20 eV)
			*m*/*z* 291→*m*/*z* 248 (FV is 135 V and CE: 18 eV)
Internal standard	Ponatinib	IS mass transitions	*m*/*z* 533→*m*/*z* 433 (FV is 145 V and CE: 18 eV)
			*m*/*z* 533→*m*/*z* 260 (FV is 140 V and CE: 20 eV)

## Data Availability

Not applicable.

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
