# Peer review of "A Liquid Chromatography Tandem Mass Spectrometry Method for the Simultaneous Estimation of the Dopamine Receptor Antagonist LE300 and Its N-methyl Metabolite in Plasma: Application to a Pharmacokinetic Study"

_molecules, 2023, doi:10.3390/molecules28041553_

Round 1
Reviewer 1 Report (Previous Reviewer 2)
How could authors modify 28- long term stability to be for LQC in three days?
Author Response
Dear Molecules Editorial Office
molecules@mdpi.com
Molecules Journal
Manuscript ID: molecules-2149331
“Editor Comments”
Please revise the manuscript according to the referees' comments and upload
the revised file within 10 days.
Please use the version of your manuscript found at the above link for your
revisions.
(I) Please check that all references are relevant to the contents of the
manuscript.
(II) Any revisions to the manuscript should be marked up using the “Track
Changes” function if you are using MS Word/LaTeX, such that any changes can
be easily viewed by the editors and reviewers.
(III) Please provide a cover letter to explain, point by point, the details
of the revisions to the manuscript and your responses to the referees’
comments.
(IV) If you found it impossible to address certain comments in the review
reports, please include an explanation in your appeal.
(V) The revised version will be sent to the editors and reviewers..
 Authors’ response
We thank the editor this opportunity to improve our manuscript and be considered again for publication in Molecules Journal. We give below detailed answers to each question raised by reviewer # 1. We mark all reply to the comments by red color in the revised manuscript.
Reviewer # 1
Comments and Suggestions for Authors
How could authors modify 28- long term stability to be for LQC in three days?
Authors’ response
We appreciate the reviewer’s his/her comments to improve our manuscript.
In stability studies, the long term stability for LQC and HQC levels of LE300 and ML300 have been done in – 80 0C and 28 days as recommended by the guidelines for bio-analytical method validation of the Food and Drug Administration US-FDA [Ref. 8].
Thank you for your kind cooperation.
Kind regards
Dr. Mohamed Attwa
Dept. Pharm. Chem.,
College of Pharmacy,
King Saud University,
P.O. Box 2457,
Riyadh, 11451,
Saudi Arabia
mzeidan@ksu.edu.sa

Reviewer 2 Report (New Reviewer)
Table 7 is missing note (a)
Suggest to include a chromatogram of actual sample
Author Response
Dear Molecules Editorial Office
molecules@mdpi.com
Molecules Journal
Manuscript ID: molecules-2149331
“Editor Comments”
Please revise the manuscript according to the referees' comments and upload
the revised file within 10 days.
Please use the version of your manuscript found at the above link for your
revisions.
(I) Please check that all references are relevant to the contents of the
manuscript.
(II) Any revisions to the manuscript should be marked up using the “Track
Changes” function if you are using MS Word/LaTeX, such that any changes can
be easily viewed by the editors and reviewers.
(III) Please provide a cover letter to explain, point by point, the details
of the revisions to the manuscript and your responses to the referees’
comments.
(IV) If you found it impossible to address certain comments in the review
reports, please include an explanation in your appeal.
(V) The revised version will be sent to the editors and reviewers..
 Authors’ response
We thank the editor this opportunity to improve our manuscript and be considered again for publication in Molecules Journal. We give below detailed answers to each question raised by reviewer # 1. We mark all reply to the comments by red color in the revised manuscript.
Reviewer # 2
Comments and Suggestions for Authors
Point # 1
Table 7 is missing note (a)
Authors’ response
We appreciate the reviewer’s his/her observation to improve our manuscript.
The note a was added to the table 7.
a Mean based on n = 6.
Point # 2
Suggest to include a chromatogram of actual sample
Authors’ response
We appreciate the reviewer’s his/her comments to improve our manuscript.
The chromatogram of actual sample has been added in the revised manuscript with the following caption:
Figure 4. Typical multiple reaction monitoring (MRM) chromatograms for in vivo rat plasma sample 2.0 h after oral (gavage) administration of 10 mg/kg LE300; LE300 (2.01 min), KLE300 (2.96 min) and IS (3.86 min).
Furthermore, Figure 4 became 5 in section 3.3. and in figure caption.
Thank you for your kind cooperation.
Kind regards
Dr. Mohamed Attwa
Dept. Pharm. Chem.,
College of Pharmacy,
King Saud University,
P.O. Box 2457,
Riyadh, 11451,
Saudi Arabia
mzeidan@ksu.edu.sa

Reviewer 3 Report (New Reviewer)
The authors in the manuscript Molecules_2149331 developed and validated an LC-MS/MS method for determination of the dopamine receptor antagonist LE300 and its metabolite MLE300 in the rat plasma. Then, the method was used to determine these compounds in a small pharmacokinetic study of LE300, when the drug was administered to rats.
The manuscript is written clearly, and the results are sufficiently explained and discussed. I have a few suggestions for improvement of the manuscript that are listed below, but I certainly think this manuscript should be accept with minor revision.
1. Some parts of the introduction should be changed. In the introduction, it is not usual to compare the developed method with methods found in the literature. This comparison should be in the chapter Results and discussion. Information on metabolite of LE300 should be added.
2. Figures 1B and 1C are not mentioned in the manuscript.
3. Table 2.: Characters a and b in the line LE300 mass transition are not explained.
4. Table 6.: Mean ± SD are listed in the table label and in the note below the table. But the ± values in the table are not given.
5. Chapter 3.2.4.: The authors write: „… spiked with 1 ng/mL LE300 (LLOQ), 2 ng/mL MLE300 (LLOQ), and IS (20 ng/mL) as set 1.” But concentration levels of LE300 and MLE300 in Table 7 are 3, 75, 175 and 6, 90,180 ng/mL, respectively.
Author Response
Dear Molecules Editorial Office
molecules@mdpi.com
Molecules Journal
Manuscript ID: molecules-2149331
“Editor Comments”
Please revise the manuscript according to the referees' comments and upload
the revised file within 10 days.
Please use the version of your manuscript found at the above link for your
revisions.
(I) Please check that all references are relevant to the contents of the
manuscript.
(II) Any revisions to the manuscript should be marked up using the “Track
Changes” function if you are using MS Word/LaTeX, such that any changes can
be easily viewed by the editors and reviewers.
(III) Please provide a cover letter to explain, point by point, the details
of the revisions to the manuscript and your responses to the referees’
comments.
(IV) If you found it impossible to address certain comments in the review
reports, please include an explanation in your appeal.
(V) The revised version will be sent to the editors and reviewers..
 Authors’ response
We thank the editor this opportunity to improve our manuscript and be considered again for publication in Molecules Journal. We give below detailed answers to each question raised by reviewer # 1. We mark all reply to the comments by red color in the revised manuscript.
Reviewer # 3
The authors in the manuscript Molecules_2149331 developed and validated an LC-MS/MS method for determination of the dopamine receptor antagonist LE300 and its metabolite MLE300 in the rat plasma. Then, the method was used to determine these compounds in a small pharmacokinetic study of LE300, when the drug was administered to rats.
The manuscript is written clearly, and the results are sufficiently explained and discussed. I have a few suggestions for improvement of the manuscript that are listed below, but I certainly think this manuscript should be accept with minor revision.
Comments and Suggestions for Authors
Point # 1
- Some parts of the introduction should be changed. In the introduction, it is not usual to compare the developed method with methods found in the literature. This comparison should be in the chapter Results and discussion.
Authors’ response
We appreciate the reviewer’s his/her comments to improve our manuscript.
The advantage of the new method is its importance for research and clinical studies applications. Therefore, comparing the developed method with the previously reported methods is vital to prove the study's significance.
Point # 2
Information on metabolite of LE300 should be added.
Authors’ response
The following paragraph has been added in the introduction section in the revised manuscript.
“Methylation is a relatively minor conjugation pathway in drug metabolism, but it is very important in the biosynthesis of endogenous compounds such as epinephrine and melatonin; and in the catabolism of biogenic amines such as dopamine, serotonin, and histamine. In many cases, this conjugation results in compounds with decreased biological activity [5]. Methylation is a two-step process. First the methyl-transferring coenzyme, S-adenosylmethionine (SAM), is biosynthesized mostly from methionine in a reaction catalyzed by methionine adenosyltransferase. Then the SAM is utilized in the transfer of the activated methyl group to the acceptor molecule (LE300) as shown in Figure 2. Heterocyclic nitrogen atoms molecule such as LE300 are susceptible to N-methylation in human results in compounds with decreased biological activity.”
Figure 2. Metabolism pathway of LE300
Point # 3
- Figures 1B and 1C are not mentioned in the manuscript.
Authors’ response
We appreciate the reviewer’s his/her comments to improve our manuscript.
We added the citation of figure 1B and figure 1C in the revised manuscript.
Point # 4
- Table 2.: Characters aand bin the line LE300 mass transition are not explained.
Authors’ response
We appreciate the reviewer’s his/her comments to improve our manuscript.
a and b characters were removed from table 2 as these were explained in the table to be FV: Fragmentor voltage and CE: Collison energy.
Point # 5
- Table 6.: Mean ± SD are listed in the table label and in the note below the table. But the ± values in the table are not given.
Authors’ response
We corrected the table label and the note below to be mean (n=6) similar to data inside the table.
Point # 6
- Chapter 3.2.4.: The authors write: „… spiked with 1 ng/mL LE300 (LLOQ), 2 ng/mL MLE300 (LLOQ), and IS (20 ng/mL) as set 1.” But concentration levels of LE300 and MLE300 in Table 7 are 3, 75, 175 and 6, 90,180 ng/mL, respectively.
Authors’ response
We corrected the typo mistakes. The table 7 is correct. We corrected the text in section 3.2.4. in the revised manuscript as follow;
“3 ng/mL LE300 (LQC), 75 ng/mL LE300 (MQC) and 175 ng/mL LE300 (HQC) for LE300, 6 ng/mL MLE300 (LQC), 90 ng/mL MLE300 (MQC), 180 ng/mL MLE300 (HQC) for MLE300”
Thank you for your kind cooperation.
Kind regards
Dr. Mohamed Attwa
Dept. Pharm. Chem.,
College of Pharmacy,
King Saud University,
P.O. Box 2457,
Riyadh, 11451,
Saudi Arabia
mzeidan@ksu.edu.sa

Round 2
Reviewer 1 Report (Previous Reviewer 2)
The editorial office is recommended to check the RAW DATA (not pdf) of the study as authors' response is not convincing to me why did not they submit the data of LQC and HQC in the first place? If you checked them then consider accepting the manuscript.
Author Response
Dear reviewer,
Please find the requested raw files for the requested data. I have also attached a screenshot for the software while opening the data and also overlayed two data from LQC and HQC. All data are available upon your request. It was a mistake from the author who wrote the manuscript that he missed these data from the first draft, sorry for any inconvenience.
With my best regards:
Mohamed W. Attwa

Round 3
Reviewer 1 Report (Previous Reviewer 2)
You may accept the article in its current status.
This manuscript is a resubmission of an earlier submission. The following is a list of the peer review reports and author responses from that submission.
Round 1
Reviewer 1 Report
A sensitive and fast LC-MS/MS method has been established and validated for the simultaneous analysis of LE300 and its N-methyl metabolite, MLE300, in rat plasma with an application in a pharmacokinetic study. The analytical plan and validation work were carried out according to the international guidelines (FDA and ICH Guidance). However, the current work is simple and routine, so I would not recommend it to be published in Molecules. And there are some concerns as follows:
1. In “2.6 Application to Pharmacokinetic Studies”, four groups of four rats each were involved in the study. Why did you use four groups of rats? For different dosage? And n=4 each group is not usually enough for rat pharmacokinetic study. The dosing route is intraperitoneally injection. Why? As you know, intravenous injection and oral gavage are usually utilized for the pharmacokinetic study, which are consistent with the clinical use. Also please explain why you chose the dosage 10 mg/kg. Why did you use the vehicle 1:1 mixture of DMSO and 0.9% saline? Too many DMSO. Did the authors confirm the concentration of dosing solution?
2. Did the authors examine different anticoagulants such as heparin, EDTA and citrate in their studies? As you know, these anticoagulants would give different effects in the LCMS/MS method.
3. The authors should further clarify the technical reasons to select ponatinib as internal standard for this analytical method.
Author Response
Dear Tacy Zhang
Assistant Editor, MDPI Tianjin
Molecules Journal
Manuscript ID: molecules-2067044
“Editor Comments”
Please revise the manuscript according to the referees' comments and upload
the revised file within 10 days.
Please use the version of your manuscript found at the above link for your
revisions.
(I) Please check that all references are relevant to the contents of the
manuscript.
(II) Any revisions to the manuscript should be marked up using the “Track
Changes” function if you are using MS Word/LaTeX, such that any changes can
be easily viewed by the editors and reviewers.
(III) Please provide a cover letter to explain, point by point, the details
of the revisions to the manuscript and your responses to the referees’
comments.
(IV) If you found it impossible to address certain comments in the review
reports, please include an explanation in your appeal.
(V) The revised version will be sent to the editors and reviewers.
 Authors’ response
We thank the editor this opportunity to improve our manuscript and be considered again for publication in Molecules Journal. We give below detailed answers to each question raised by reviewer # 1. We mark all reply to the comments by red color in the revised manuscript.
Reviewer # 1
Comments and Suggestions for Authors
A sensitive and fast LC-MS/MS method has been established and validated for the simultaneous analysis of LE300 and its N-methyl metabolite, MLE300, in rat plasma with an application in a pharmacokinetic study. The analytical plan and validation work were carried out according to the international guidelines (FDA Guidance). However, the current work is simple and routine, so I would not recommend it to be published in Molecules. And there are some concerns as follows:
Authors’ response
We appreciate the reviewer’s words and his/her suggestions to improve our manuscript. We give below our answer to his/her concerns.
Point # 1
- In “2.6 Application to Pharmacokinetic Studies”, four groups of four rats each were involved in the study. Why did you use four groups of rats? For different dosage? And n=4 each group is not usually enough for rat pharmacokinetic study. The dosing route is intraperitoneally injection. Why? As you know, intravenous injection and oral gavage are usually utilized for the pharmacokinetic study, which are consistent with the clinical use.
Authors’ response
According to the reviwer comments, we re-study the protocol of pharmacokinetic study for LE300 according to the Approval of the study by the Animal Ethics Committee of pharmacology department at the College of Pharmacy in King Saud University, Saudi Arabia, and validated the results of the work. The following paragraph have been added in the revised manuscript. Section 2.6.
“Six healthy Wistar male rats (250 ± 30 g) were kindly gifted from the Experimental Animal Care Center. The animals were maintained in a well-ventilated room inside cages under a 12-hr day/night cycle at specific conditions (40%–60% relative humidity and 24–27 0C temperature). All the rats had free access to water while the diet was stopped for 12 hr prior to drug loading. The rats were kept for one week in a laboratory before performing the experiments. On the day of experiments, rats were treated by gavage administration with a single oral dose of 10 mg/kg LE300 dissolved in 1% DMSO/saline. [16]. Blood samples (300 μL) were collected into heparinized 1.5 mL polythene tubes containing ethylenediamine tetraacetic acid dipotassium (EDTA K2) (anticoagulant) before drug administration, and at 0.5, 1, 2, 4, 6, 8, 12, 18, and 24 h after oral administration of a single oral dose of 10 mg/kg LE300 [16]. The samples were directly centrifuged at 3000 rpm for 10 min at 4 °C. The plasma obtained was stored at −80 °C until analysis.”
Point # 2
Also please explain why you chose the dosage 10 mg/kg.
Authors’ response
We appreciate the reviewer’s his/her suggestions to improve our manuscript.
According to Ref. # 16, rats were treated by gavage administration with a single oral dose of 10 mg/kg LE300 dissolved in 1% DMSO/saline [16].
Point # 3
- Did the authors examine different anticoagulants such as heparin, EDTA and citrate in their studies? As you know, these anticoagulants would give different effects in the LC-MS/MS method.
Authors’ response
We collected the blood samples into heparinized 1.5 mL polythene tubes containing ethylenediamine tetraacetic acid dipotassium (EDTA K2) as anticoagulant. The EDTA K2 (Inorganic compound) does not have effect in the LC-MS/MS method.
Point # 4
- The authors should further clarify the technical reasons to select ponatinib as internal standard for this analytical method.
Authors’ response
We appreciate the reviewer’s comment and his/her suggestions to improve our manuscript.
We investigated the use of different internal standards, such as pemigatinib, nateglinide, repaglinide, chloroquine, and hydroxychloroquine, but such internal standards either gave poor peaks or led to overlapping with LE300 or MLE300. Ponatinib was selected as the method’s IS, whereas it has a higher extraction recovery (≥ 99%) and performance characteristics to LE300 and MLE300 [8].
This paragraph has been added in the revised manuscript.
Thank you for your kind cooperation.
Kind regards
Dr. Mohamed Attwa
Dept. Pharm. Chem.,
College of Pharmacy,
King Saud University,
P.O. Box 2457,
Riyadh, 11451,
Saudi Arabia
mzeidan@ksu.edu.sa
Reviewer 2 Report
The paper describes the LC-MS/MS method for simultaneous determination dopamine receptor antagonist LE300 and its N-methyl metabolite in rat plasma.
The manuscript is well written and organized. However, the authors should address the following major comments:
1- The internal standard (IS) should be either a stable isotopically labelled or structure analogue of the analyte. Ponatinib is not suitable as an internal standard for the current method as can be seen in figure 3 where its retention time is different from either analytes.
2- Low quality control sample should be within 3 X of LLOQ.
3- Recovery of IS should be provided.
4- Accuracy and precision should be done on LLOQ, LQC, MQC and HQC samples.
5- Levels used in stability study should be LQC and HQC.
6- In section 3.1.: 42% can should be 42% ACN.
Author Response
Dear Tacy Zhang
Assistant Editor, MDPI Tianjin
Molecules Journal
Manuscript ID: molecules-2067044
“Editor Comments”
Please revise the manuscript according to the referees' comments and upload
the revised file within 10 days.
Please use the version of your manuscript found at the above link for your
revisions.
(I) Please check that all references are relevant to the contents of the
manuscript.
(II) Any revisions to the manuscript should be marked up using the “Track
Changes” function if you are using MS Word/LaTeX, such that any changes can
be easily viewed by the editors and reviewers.
(III) Please provide a cover letter to explain, point by point, the details
of the revisions to the manuscript and your responses to the referees’
comments.
(IV) If you found it impossible to address certain comments in the review
reports, please include an explanation in your appeal.
(V) The revised version will be sent to the editors and reviewers.
 Authors’ response
We thank the editor this opportunity to improve our manuscript and be considered again for publication in Molecules Journal. We give below detailed answers to each question raised by reviewer # 2. We mark all reply to the comments by red color in the revised manuscript.
Reviewer # 2
Comments and Suggestions for Authors
The paper describes the LC-MS/MS method for simultaneous determination dopamine receptor antagonist LE300 and its N-methyl metabolite in rat plasma.
The manuscript is well written and organized. However, the authors should address the following major comments:
Authors’ response
We appreciate the reviewer’s words and his/her suggestions to improve our manuscript. We give below our answer to his/her concerns.
Point # 1
1- The internal standard (IS) should be either a stable isotopically labelled or structure analogue of the analyte. Ponatinib is not suitable as an internal standard for the current method as can be seen in figure 3 where its retention time is different from either analytes.
Authors’ response
We appreciate the reviewer’s comment and his/her suggestions to improve our manuscript.
The applied of stable isotopically labelled internal standard (IS) is recommended from US-FDA guidelines [8], Otherwise, it is high expensive to purchage it.
“We investigated the use of different internal standards, such as pemigatinib, nateglinide, repaglinide, chloroquine, and hydroxychloroquine, but such internal standards either gave poor peaks or led to overlapping with LE300 or MLE300. Ponatinib was selected as the method’s IS, whereas it has a higher extraction recovery (≥ 99%) and performance characteristics to LE300 and MLE300 [8].”
This paragraph has been added in the revised manuscript.
Furthermore, Figure 3 shows overlayed chromatograms with good resolution (Rs ≥1.5) for LE300, MLE300, and IS in rat plasma, which is importance in chromatography separations.
Point # 2
2- Low quality control sample should be within 3 X of LLOQ.
Authors’ response
We appreciate the reviewer’s his/her suggestions to improve our manuscript.
The following paragrapg has been added in the revised manuscript, section 2.3.
“Different quality control samples for LE300 at 1.0 ng/mL for the LLOQ, 3.0 ng/mL for the QC sample at low concentration, (LOQ), 75 ng/mL for the QC sample at mid concentration, (MQC) and 175 ng/mL for the QC sample at high concentration (HQC) were prepared by spiking appropriate volume of the intermediate solutions with blank rat plasma. Similarly, different quality control samples for MLE300 at 2.0, 6.0, 90 and 180 ng/mL for LLOQ, LOQ, MQC and HQC were prepared.”
Point # 3
3- Recovery of IS should be provided.
Authors’ response
Recovery of IS has been added in Table 7, in the revised manuscript.
Point # 4
4- Accuracy and precision should be done on LLOQ, LQC, MQC and HQC samples.
Authors’ response
We appreciate the reviewer’s his/her suggestions to improve our manuscript.
In Table 6 and section 3.2.3. in the revised manuscript, accuracy and precision have been done on four quality control levels, LLOQ, LQC, MQC and HQC.
Point # 5
5- Levels used in stability study should be LQC and HQC.
Authors’ response
In Table 8 and section 3.2.5. in the revised manuscript, stability study have been done on LQC and HQC.
Point # 6
6- In section 3.1.: 42% can should be 42% ACN.
Authors’ response
The correction done.
Thank you for your kind cooperation.
Kind regards
Prof. Mohamed Hefnawy,
Dept. Pharm. Chem.,
College of Pharmacy,
King Saud University,
P.O. Box 2457,
Riyadh, 11451,
Saudi Arabia
Round 2
Reviewer 1 Report
On the basis of my opinions in the first review, the author has made some modifications and improvements to a certain extent. However, in terms of innovation and scientific importance, the current research cannot be included in those excellent papers selected by Molecules. Therefore, I regret that I cannot recommend its publication. And there are my concerns as follows: In the revised manuscript “2.6 Application to Pharmacokinetic Studies”, the dosing route has been revised from “intraperitoneally injection” into “oral dose”. But in Figure 4 and Table 9, the dosing route is still intraperitoneally injection. Major mistakes and inconsistent statements in your manuscript. Therefore, the readers have reason to doubt the authenticity and reliability of the full-text data. Also please explain why you chose the dosage 10 mg/kg. Why did you use the vehicle 1:1 mixture of DMSO and 0.9% saline? Too many DMSO. Did the authors confirm the concentration of dosing solution?".
This paper gives some unbelievable data, although I don't think it is the authors' intention. So I have to reject it from publication in your Journal.
Reviewer 2 Report
The authors replied to my comment concerning stability study " Levels used in stability study should be LQC and HQC" as follows: In Table 8 and section 3.2.5. in the revised manuscript, stability study have been done on LQC and HQC.
Meanwhile, the level used is 1 ng/mL which represents LLOQ not LQC? Please explain.